# A Peculiar Binding Characterization of DNA (RNA) Nucleobases at MoOS-Based Janus Biosensor: Dissimilar Facets Role on Selectivity and Sensitivity

**DOI:** 10.3390/bios12070442

**Published:** 2022-06-23

**Authors:** Slimane Laref, Bin Wang, Sahika Inal, Salah Al-Ghamdi, Xin Gao, Takashi Gojobori

**Affiliations:** 1Computational Bioscience Research Center (CBRC), King Abdullah University of Science and Technology (KAUST), Thuwal 23955-6900, Saudi Arabia; sahika.inal@kaust.edu.sa (S.I.); xin.gao@kaust.edu.sa (X.G.); takashi.gojobori@kaust.edu.sa (T.G.); 2School of Chemical, Biological and Materials Engineering, Center for Interfacial Reaction Engineering (CIRE), University of Oklahoma, Norman, OK 73019, USA; wang_cbme@ou.edu; 3Biological and Environmental Science and Engineering (BESE), King Abdullah University of Science and Technology (KAUST), Thuwal 23955-6900, Saudi Arabia; 4Physics Department, Faculty of Science, University of Tabuk, Tabuk 71451, Saudi Arabia; saalghamdi@ut.edu.sa

**Keywords:** Janus material, 2D layer, DFT, molecular states, nucleobases, DNA/RNA

## Abstract

Distinctive properties of Janus monolayer have drawn much interest in biotechnology applications. For this purpose, it has explored theoretically all sensing possibilities of nucleobases molecules (DNA/RNA) by Janus MoOS monolayer on both oxygen and sulfur terminations by means of rigorous first–principles calculation. Indeed, differences in interaction energy between nucleobases indicate that a monolayer can be used for DNA sequencing. Exothermic interaction energy range for DNA/RNA molecules with both oxygen and sulfur sides of the Janus MoOS surfaces have been found to range between (0.61–0.91 eV), and (0.63–0.88 eV), respectively, and the binding distances indicate that these molecules bind to both facets by physisorption. The exchange of weak electronic charges between the MoOS monolayer and the nucleobases molecules has been studied by means of Hirshfeld-I charge analysis. It has been observed that the introduction of DNA/RNA nucleobases molecules alters the electronic properties of both oxygen and sulfur atomic layers of the Janus MoOS complex systems as determined by plotting the 3D Kohn–Sham frontier orbitals. A good correlation has been found between the interaction energy, van der Waals energy, Hirshfeld-I, and *d*–band center as a function of the nucleobase’s affinity, and the interaction energy, suggesting adsorption dominated by van der Waals interactions driven by molybdenum *d*–orbital. Moreover, the lowering in the adsorption energy leads to an active interaction of the DNA/RNA with the surfaces, accordingly its conduct to shorter the recovery time. The selectivity of the biosensor modulation device has illustrated a significant sensitivity for the nucleobases on both the oxygen and sulfur layer sides of the MoOS monolayer. This finding reveals that apart from graphene, dichalcogenides–Janus transition metal may also be adequate for identifying DNA/RNA bases in applied biotechnology.

## 1. Introduction

Selective recognition of diverse types and sequences of deoxyribonucleic and ribonucleic acids (DNA and RNA) present in the human body is crucial, and efficient nanosensors for daily applications are of vital importance. DNA (RNA) is a double (single) twisted, staircase-like helix molecule, frequently called the *backbone*. These base molecules are in sequence with the deoxyribose (ribose) helix of phosphate–sugar bonds where every single termination of the staircase contains two hydrogen-bonded molecules named nucleobases. Although nucleobases are principally the building blocks of the amino acid and sequence form peptides and proteins, four distinct nucleobases are found in DNA (RNA) bases: guanine (G), adenine (A), cytosine (C), and thymine (T) (uracil (U)), respectively. Meanwhile, all DNA/RNA are based on aromatic heterocyclic organic rings. DNA and RNA play a crucial role in all classes of metabolisms and related processes and their biosensing and nucleobases sequence identification represent key studies in the prospect of the applied biomedical domain [1,2,3]. A nucleobase sensor, apart from being able to identify diverse components under in vivo conditions, would be hydrophilic and compatible with the cytosol medium in cells. Sensing is a surface phenomenon that promotes the attraction of a two-dimensional (2D) device in contact with the DNA/RNA. Particularly, it leads to the possibility of identifying a small concentration of molecules with 2D materials [4,5,6].

Two-dimensional (2D) layers have attracted great consideration due to their typical sandwich properties [7,8,9]. Janus–type forms with transition metal dichalcogenides (TMDs) were reported. In the 2D crystals of TMDs, the transition metal atoms are sandwiched between chalcogen atoms forming MX_2_ composition (M = Mo, W while X = S, Se, Te), and have been fascinating for being atomically ultrathin quantum emitters [8,9]. Motivated by the great success of graphene, many 2D materials have been developed to serve numerous nanotechnology applications such as electronic devices, environmental bioscience, food, drug vehicle therapy, and biomedicine [10,11,12]. As an example, transition metal dichalcogenides (TMDs), carbonitrides (MXenes), hexagonal boron nitride (h–BN), and their derivatives are realized with few layers that exhibit a weak out-of-plane van der Waals interlayer bonding and a strong in-plane covalent bonding [13,14,15,16,17,18]. Janus transition metal dichalcogenides (TMDs) have become a hot research topic in the latest few years due to their unique physical properties introduced by the asymmetry dissimilar elementary assembly [19,20,21]. MoSSe single-layer has been extensively investigated both experimentally and theoretically because of its distinctive structural, electronic, mechanical, and optical properties [22,23,24,25,26]. The electronic nature of 1H–MoS_2_ has made it a suitable candidate for applications in biosensing [7]. Earlier experimental investigations have demonstrated that MoS_2_ can be synthesized by chemical vapor deposition (CVD), where MoO_3_ could be used as the Mo precursor [27,28,29]. Furthermore, it has also been revealed that incomplete sulfurization of MoO_3_ leads to the realization of either MoO_3−x_S_y_, molybdenum oxysulfides, or MoO_3−x_, molybdenum oxides [30,31,32,33]. Therefore, the formation of an out-of-plane asymmetric MoOS monolayer can be achieved in such experiments. Many different Janus TMDs have been predicted theoretically and only Janus 1H–MoSSe has been successfully fabricated experimentally [34,35].

The charge distributions of the O–Mo layer and the S–Mo layer are significantly different due to differences in the electronic arrangements (the electro-negativities and atomic radius of O and S are quite dissimilar) [36]. Correspondingly, the formation process of Janus MoOS monolayer is basically associated with the synthesis of single-layer MoS_2_ from MoO_3_ powders, which motivated us to predict such a possible new biosensor 1H phase of Janus MoOS composed of O–Mo–S atomic layers [31,32,33]. Alternatively, since the formation of molybdenum oxysulfides, MoO_3−x_S_y_, has been observed in previous experiments, it may also be possible to construct a Janus monolayer in such an experiment as an intermediate state. Accordingly, an inner electric field perpendicular to the surface is built in Janus MoOS monolayer [36]. The interior electric field is assumed to have an analogous effect similar to one produced by an external electric–field formed by the gate bias. An additional gate bias will make sensing devices complex and power-wasting. Janus MoOS monolayer with an interior electric field exhibits a high sensitivity potential and flexible modulation without gate bias.

For this purpose, it has been shown for this new biosensor has two different termination facets joined by nucleobases molecules located on their dissimilar sides revealing that DNA/RNA on the MoOS-based Janus monolayer represents an active potential biosensor. In this work, the interaction characteristics and thermodynamic properties at vacuum and implicit solvent levels for deoxyribonucleic and ribonucleic acid molecules on a 2D Janus MoOS monolayer have been studied. By means of density functional theory (DFT), the binding characteristics, selectivity, and sensitivity have been unveiled crossbreed for biomolecules on the 1H-MoOS Janus system.

## 2. Methodology

The DFT calculations were carried out using the Vienna Ab-initio Simulation Package (VASP 5.4) [37,38]. All electrons with projected augmented wave (PAW) formalism were used to model the electron–ion interactions [39]. The exchange-correlation functional contribution to the total energy was modeled by the generalized gradient approximation (GGA). In addition to the modified Perdew–Burke–Ernzerhof (PBE) for treating the van der Waals (vdW) interaction corrections [40], the dispersion correction with Beck–Johnson (D3–BJ) approach was presented in all calculations [41,42], as described by Grimme et al. [43,44]. A supercell model of 4 × 4 boxes was used, and a vacuum space of 20 Å was applied to avoid the interaction between replicated layers along the z-direction. At reciprocal space integration, a mesh of 4 × 4 × 1 Γ-centered point was applied for geometry relaxations, while 8 × 8 × 1 was set for electronic properties. A total energy convergence criterion of 1 × 10^−6^ eV was employed for the self–consistent solution of the Kohn–Sham (KS) equations. The pseudo-wave functions were expanded in a plane-wave basis set with an energy cut-off of 400 eV. The geometry was relaxed until the residual force is smaller than 0.01 eV Å^−1^.

Adsorption energies *E*_Ads_ of nucleobases on MoOS–Janus-based 2D materials were calculated by subtracting the energies of DNA(RNA) molecules and the MoOS from the energy of the slab, as shown in the following formula:*E*_Ads_ = −*E*_MoOS_ − *E*_DNA(RNA)_ + *E*_MoOS/DNA(RNA)_(1)
where *E*_MoOS_, *E*_DNA(RNA)_, and *E*_MoOS/DNA(RNA)_ represent the DFT-calculated energies of the clean 2D MoOS material, the nucleobases biomolecules, and the total system including the adsorbed biomolecule and Janus material, respectively. The total adsorption energies can be further decomposed into the electronic energy and dispersion energy, which are calculated by Grimme’s DFT-D3 method [41].

Ideally, the nucleobase biomolecules are released from the surface at an external stimulus such as rising temperatures; this process is simulated by calculating the Gibbs free energy of the complex with and without the DNA(RNA). The Gibbs free energy changes for all intermediate configurations during the loading and releasing process are calculated as follow:Δ*G* = Δ*E_DFT_* + Δ*E_ZPE_* + *T* Δ*S*(2)
where Δ*E_DFT_*, Δ*E_ZPE_*, and *T* Δ*S* represent the difference of DFT–calculated adsorption energy, zero-point energy (ZPE), and entropy contribution, respectively.

## 3. Results and Discussions

The MoOS–Janus, constructed within three-fold coordination of S–Mo–O atomic layers, is asymmetric dichalcogenides 2D monolayer facets of 1H–MoOS phase, the main dynamically stable geometry of the Janus system with a hexagonal lattice [36]. Regarding the structural composition of the 1H–MoOS surface, the Mo atomic layer is sandwiched between the O and S layers with Mo covalently bonded to O and S atoms. The corresponding in-plane lattice constant of the MoOS single–layer is calculated as *a* = *b* = 2.99 Å. Thereby, Mo–S and Mo–O bond lengths are 2.38 and 2.08 Å, respectively, which are in good agreement with previously reported data [36]. In order to determine the adsorption geometries of G, C, A, T, and U biomolecules on both sides of the 1H–MoOS monolayer, the nucleobase molecule is placed parallel to the top of a 4 × 4 supercell of the MoOS surface sides. In addition, the whole system is fully relaxed. Several possible adsorption sites and typical orientations have been considered. Obviously, the top site above the center of the hexagon, the top of the Mo (O or S) atom, and the top site above the Mo–O (Mo–S) bond are incorporated, while configurations of the biomolecules are being parallel to the monolayer surfaces. The most favorable configurations of the nucleobase molecules adsorbed on the O layer, and S layer of the Janus MoOS monolayer are presented in Figure 1. Regardless, the structural geometry of the relaxed nucleobases species on Janus MoOS stays parallel to both sides of the surface, while the others, specifically T, are slightly tilted.

It is found also that in all configurations, the center of pyrimidines (C_4_H_4_N_2_) aromatic ring adsorbs similarly on the uppermost of the oxygen (sulfur) atomic layer. For A and G molecules, the second ring attempts to align its center with nearby oxygen (sulfur) of the top monolayer. On the other hand, the aromatic ring of the U molecule is remarkably aligned with the O layer (S layer), revealing the most symmetry nucleobase overall. Further, the atom next to the O layer (S layer) termination is a hydrogen (H), either from an amino (NH_2_) or a methylene (CH_2_) group, whenever these are found, or an oxygen (O) atom, in the case of U. The adsorption takes place with the nearest atom sitting more than 2.5 Å (3.0 Å) away from the O layer (S layer) of the Janus MoOS surface and presents tiny unseen structural deformations, weaker than van der Waals (vdW) interactions. Further, the optimized structures for adsorbed G, A, C, T, and U on the O layer are very similar to the corresponding ones on the S layer (see Figure 1). Table 1 listed the vertical distance between the nucleobase molecules and 1H–MoOS monolayer, obtained using PBE−D3(BJ) functional. As illustrated in Figure 1, the calculated vertical distance range is 2.74–2.94 Å for the O layer side, and 3.06–3.19 Å for the S layer termination side, respectively. The adsorption configurations of G, C, A, T, and U on Janus MoOS are nearly the same as those on monolayer MoO_2_ and MoS_2_, as seen in Figure 1.

Table 1 summarizes the calculated energy results, including nearest molecular–monolayer spacing, adsorption energy, interaction energy, van der Waals energy, and the covalent energy for each biomolecule on the 1H–MoOS monolayer. Clearly, nucleobases on both sides of the 1H–MoOS monolayer energetically adopt a parallel orientation. Related to the adsorption of nucleobases on the Janus MoOS Mo–O side (Mo–S side), the order of adsorption energies is G > A > C > T > U, which is very similar to those of 2D pristine MoS_2_ [45,46,47]. The adsorption energies of A, C, T, and U on the O layer (S layer) are −0.86 (−0.89), −0.81 (−0.82), −0.71 (−0.76), and −0.64 (−0.67) eV, respectively. For G, however, the adsorption energy reaches up to −0.90 (−0.91) eV, which is noticeably higher than those of the other four biomolecules. However, the adsorption distances of G on the O layer (S layer) are nearly 2.77 (3.15) Å, whereas, for A, C, T, and U are 2.80 (3.18), 2.74 (3.06), 2.94 (3.19), and 2.85 (3.19) Å, respectively. Interestingly, the simultaneous interaction between the two terminated surfaces and biomolecules tends to be reduced over the Janus MoOS monolayer to minimize the vdW forces, for more detail see Table 1 and Figure 1. Implicitly, this also could be unveiled by the interaction energy of standard GGA–PBE (covalently) energy term that is less predominant in all molecular orientations at the surfaces. The differences in adsorption energy and adsorption distance show that sensitivity between 1H–MoOS surfaces and G is higher than those of A, T, C, and U, respectively. The adsorption energy values are all negative, demonstrating thermodynamically favorable adsorption characteristics. In Table 1, the moderate binding energies are close to −1 eV for the nucleobases on either O or S atomic layer sides of the 1H–MoOS monolayer, suggesting that the molecules are physisorbed.

In addition, by exploring the energy decomposition (see Table 1 and Table 2), the binding energy is split into destabilized (positive) and minor deformation energies related to G, A, C, T, and U adsorption states, respectively, and stabilized (negative) with predominant interaction energies between nucleobases with oxygen and sulfur layer sides of the Janus MoOS surfaces (vdW interactions). In addition, the reason that G exhibits the largest binding energy or smallest interaction energy in both sides (O layer or S layer) of the Janus MoOS monolayer and other biomaterials [7,18,45,48,49,50,51,52], is relatively associated with high polarizability prior to other nucleobases [50,52,53]. This trend is more encouraging by the charge density distributions and molecular interactions (in the Appendix A). In contrast, with other reported data regarding DNA bases, except U, on other substrates, the interaction energies remain to be improved onto the O layer (S layer) of the Janus MoOS in the range of 5–15% for phosphorene [49], 17–25% in the case of *h*–BN [50], and finally 15–21% onto graphene [52]. These results qualitatively illustrate the enhanced interaction strength on both sides of the Janus 1H–MoOS complex systems. Therefore, MoOS monolayer is primly appropriate for physisorption of nucleobases than graphene, tellurene, phosphorene, and boron nitride nanosheets. For the purpose of comparison, the parallel adsorption calculations are repeated, including an implicit solvation model, better mimicking realistic conditions, as implemented in VASPsol [54,55]. The resulting adsorption configurations remained practically unchanged, while the adsorption became weaker by ~0.10–0.20 eV overall (see Appendix A). Moreover, on the basis of the Hirshfeld-I charge analysis, Figure 2 shows that the biomolecules assist on trivial *n*–dopants and bring–in up to 0.06(0.13)–0.15(0.19) electron/molecule in the O layer (S layer) MoOS monolayer. At an ambient temperature of 298 K and ambient pressure of one bar, the Gibbs free energies of the adsorption states are also negative for all geometry configurations, while with a solvent, the adsorption still remains exothermic for the S layer, other than the U case, and could be endothermic by up to 1.5 meV Å^−2^ for the O layer, as illustrated in Figure 3. It is worth noting that the highest lower orbitals (HOMO–LUMO) gap has an important dependency on the binding energy (see Appendix A).

To understand the effect of the adsorption and sensing process, the Kohn–Shan frontier orbital of the nucleobases/surfaces is investigated at high symmetry Γ–point by means of HSE06 functional [56]. Equally, 3D plots of the HOMO–LUMO of the surface and the ones with the strongest and weakest binding strength, respectively, with the adsorbed G or U molecule, are shown in Appendix A. The nucleobases do not influence the highest occupied molecule orbital except for the sulfur side (Appendix A), and the lowest unoccupied molecular orbital is located at the O and Mo atomic states of the monolayer. As shown in Appendix A for the oxygen adsorption side, the nature of the HOMO is basically unchanged upon the adsorption of nucleobases, which provides a fact that the geometry of adsorption systems remains almost unchanged in the adsorption process. Meanwhile, these results confirm that the adsorption of nucleobases on the O layer of the Janus MoOS is not chemical in essence. However, compared with the S layer, the HOMO of the adsorption systems has obvious changes, where the HOMO is exclusively concentrated on the Janus MoOS monolayer. This behavior is predominantly due to nucleobases with 1H–MoOS, which makes the band shift toward lower energy. Additionally, the Fermi surface level found by the nucleobase’s contribution substitutes the original energy level of nucleobases to become HOMO-1 in the adsorbed system (see Appendix A). Hence, overall, the HOMO of nucleobases–Janus complexes is principally initiated from the amino (–CH_2_) moiety, methylene (–NH_2_) group, molybdenum (Mo), sulfur (S), and/or oxygen (O) atoms of nucleobases and MoOS monolayer, which is consistent with previous results on DNA/Penta-graphene [51]. These molecular orbitals are quantitatively and qualitatively very similar, except for uracil on the sulfur termination, as anticipated for weak interaction, and equally preferred for effective preservation of the sensing surface. Further, Appendix A show the HOMO–LUMO gaps of optimized adsorption systems, which are considerably reduced related to standalone Janus MoOS. Hence, the degree of freedom reduction is associated with the adsorption regime of different nucleobase molecules, and the HOMO–LUMO gap drops as the interaction energy increases. In this aspect, 1H–MoOS is a semiconductor material, so the changes in the orbital gap could be useful for molecule recognition and sensing applications. Upon O layer physisorption, there are obvious changes of 0.89–1.97 eV in the HOMO–LUMO gap of the adsorbate–adsorbent complexes. All five nucleobases decrease the values (up to 0.89 eV for G) with respect to the isolated 1H–MoOS monolayer, except for S layer complex systems (2.18–2.19 eV), which stays close to that of the Janus MoOS (~2.3 eV) standalone. The surface with either O or S side adsorbed G molecule with the maximum interaction energy, presents the minimal value of HOMO–LUMO gap in all adsorption models.

Furthermore, the Kohn–Sham frontier orbitals of both sides of nucleobases–MoOS adsorption systems are enabled to partake in the 2*p* orbital of C, N, and/or O atoms (HOMO-1 in the case of the S layer) of all five nucleobases hybridize with the 2*p* and 4*d* orbitals of O and Mo atoms of the Janus MoOS near the Fermi level, as represented in Appendix A. No visible hybridization and/or contribution from the nucleobases is observed at the LUMO state above the Fermi level (see Appendix A). Except for –CH_2_ moiety for A, G, and C bases, –NH_2_ moiety is the main group implicated in hybridization and the O atom is basically the primary contribution atom for T and U bases near the Fermi level. The contribution of a hydrogen atom is nearly zero close to the Fermi level, and it only appears at some low energy in the HOMO state. In addition, it appears that in the O layer, a new peak in the vicinity of the U–MoOS system’s Fermi level is the reason for the increase in the HOMO–LUMO gap, which is mainly due to the *sp* hybridization between C, N and O atoms. Unlikely, the G–MoOS complex system has a smaller HOMO–LUMO gap.

Upon the outward vdW–driven adsorption and the affinity for larger nucleobases density to adsorb more strongly hints towards a possible correlation between molecular density volumes and interaction energies *E*_Int_ of adsorbed complex systems. These two properties display a squared linear correlation coefficient of R^2^ = 0.72 (0.77) for the Mo–O (Mo–S) side, as shown in Figure 4. However, in Figure 5, the vdW energy *E*_vdW_ does correlate well with molecular density volumes and has a value of R^2^ = 0.86 (0.82) for the Mo–O (Mo–S) side. Regarding *E*_Int_ and *E*_vdW_, the slope of the regression line is about 1.2 times that of *E*_vdW_, indicating a stronger effect on the adsorption strength at lower temperatures. One other fundamental quantity concerns the electronic charge transfer from the nucleobases to the Janus MoOS monolayer and the interaction energy. It is found also that the Hirshfeld-I charge correlates strongly with the interaction energy by an R^2^ = 0.90 (0.86) for Mo–O (Mo–S) side. A more quantitative analysis of the electronic transfer to the Janus MoOS (O and S sides) as well as the shifts in the Mo *d*-band is also possible, as represented in Figure 6 and Figure 7. The shifts in the Mo *d*-band upon adsorption are quantified by determining the position of the *d*-band center for the Mo atom in the Janus MoOS and its facets. The filled *d* states of all Mo atoms in the slab are used in this matter. As seen from the data in Figure 7, the *d*-band centers undergo an appreciable downward shift, relative to their average position in the unsupported nucleobases, upon binding to the Janus MoOS monolayer. Once again, there is a positive correlation of an R^2^ = 0.94 (0.93) between the strength of the binding and the downward shift of the *d*-band center in the case of the Mo–O (Mo–S) side. The electronic structure of the DNA/RNA is modified both through chemical interaction and through slight deformation induced in the surfaces. Indeed, there is a clear indication of charge transfer from the nucleobases to both Oxy/Sul supports of the Janus MoOS, accompanied by a shift of the Mo *d*-band away from the Fermi level, which increases in the presence of support sides. The interaction energy of nucleobases molecules, for example, is found to decrease as one is stimulated from G to a U one on an O layer, and to one on an S layer support; this provides a credible explanation for the increased G tolerance of the Janus MoOS monolayer.

Figure 8 illustrates the recovery time at either vacuum or solvent interfaces. It is well known that the adsorption energy and energy gap affect the sensing characters. However, strong interactions are not favorable during the adsorption process as the sensing becomes long and requires hard desorption of the DNA/RNA from the sensor. Note that the lower binding energy for the case of G implies an attractive interaction with the Janus MoOS surface on both sides. This leads to a recovery time that is shorter (further shorter in the presence of aqueous solvent) than for any other nucleobases studied here, by feature of the Boltzmann factor, i.e., *τ*(%) ∼ exp(−*E*_Ads_/k_B_T) × 100. On the other hand, the larger binding energies for T and U, which are higher than what would correspond to the thermal energy at room temperature T = 298 K indicate that thermal fluctuations may not change the structural configuration of the chemical connection in those cases. It will rather only be the exact chemical identity and configuration of the nucleobase that will distinctively influence the sides of the Janus either Mo–O or Mo–S facets, and thereby the current modulation of the device. Further, for quantitative analysis of the selectivity of the modulation sensor device, sensitivity is defined as *s*(%) ∼ exp(Δ*E*_g_/k_B_T) × 100. Figure 9 presents the *s*(%) values for adsorption of the DNA/RNA over the Janus MoOS monolayer at the vacuum (solvent) phase. It indicates that the %Δ*E*_g_ ~ 81.7% (94.7%) is the highest for G (lowest for U) on the MoOS complex O layer, whereas in the case of the S layer it decreases even with a solvent, suggesting the Janus MoOS monolayer is a promising candidate for the potential biosensor to detect the nucleobases molecules. Based on the results obtained in this study, both oxygen and sulfur atomic surface sides can be utilized to classify different nucleobases, which are in the application of protein and DNA/RNA sequencing, respectively. The interactions of deoxyribonucleic and ribonucleic acid molecules with the Janus 1H–MoOS monolayer would indicate also that both surface facets can be beneficial to recognize the heterogeneous catalysis and molecules’ complex crystallization.

## 4. Conclusions

By using the state-of-the-art DFT–D3-based approach, the adsorption, interaction, deformation, Kohn–Sham orbital, and Gibbs free energy are carried out in vacuum and solvent phases. The results obtained on both oxygen and sulfur sides shows that monolayer MoOS exhibits great potential applications in DNA/RNA sensing performance due to the obvious differences in recovery time and sensitivity parameters. These could be utilized to characterize different nucleobases molecules, and thus would be applied in biotechnology such as in the determination of DNA/RNA sequencing. The interaction, vdW energy, Hirshfeld-I charge, and *d*-band center display a proportional correlation with the molecular affinity volumes of the biomolecules on 2D monolayer MoOS complex surfaces. This, along with the small Hirshfeld-I charge differences, between 0.06(0.13)–0.15(0.19) electron/molecule in the O layer (S layer), suggests physisorption with a small, and obvious, charge transfer. The outcomes of Kohn–Sham frontier orbitals also indicate that nucleobase molecules are physisorbed on the Janus MoOS monolayer throughout the weak forces of vdW interaction. The charge density shows that orbital hybridization is the primary cause of charge transfer between biomolecules and the Janus MoOS monolayer in both facets. In addition, the lower binding energy point toward an attractive interaction with the Janus MoOS surface on both sides, thus leading to a shorter recovery time. The selectivity of the modulation sensor device shows higher (lower) sensitivity for G (U) bases on either O layer or S layer sides. In summary, it has been revealed that Janus MoOS monolayer with either oxygen or sulfur facets would be an encouraging candidate for biosensors to detect DNA/RNA nucleobases molecules.

## Figures and Tables

**Figure 1 biosensors-12-00442-f001:**
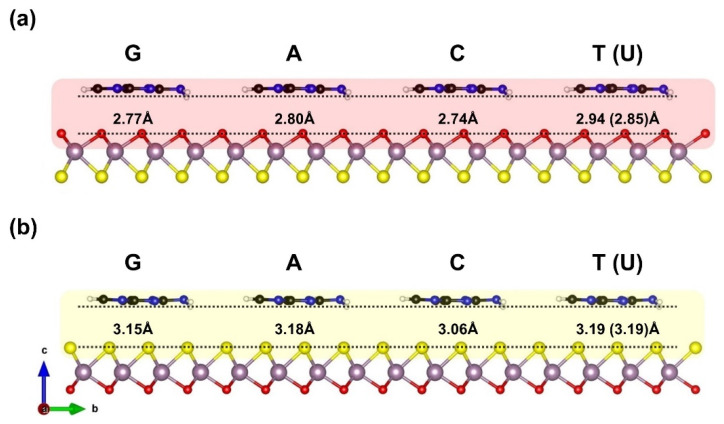
Adsorption configurations model and optimized nucleobases molecules distance above the Janus MoOS monolayer facets, (**a**) on O layer, (**b**) on S layer, respectively. Color code; purple: Mo, red: O, yellow: S, black: C, blue: N, white: H.

**Figure 2 biosensors-12-00442-f002:**
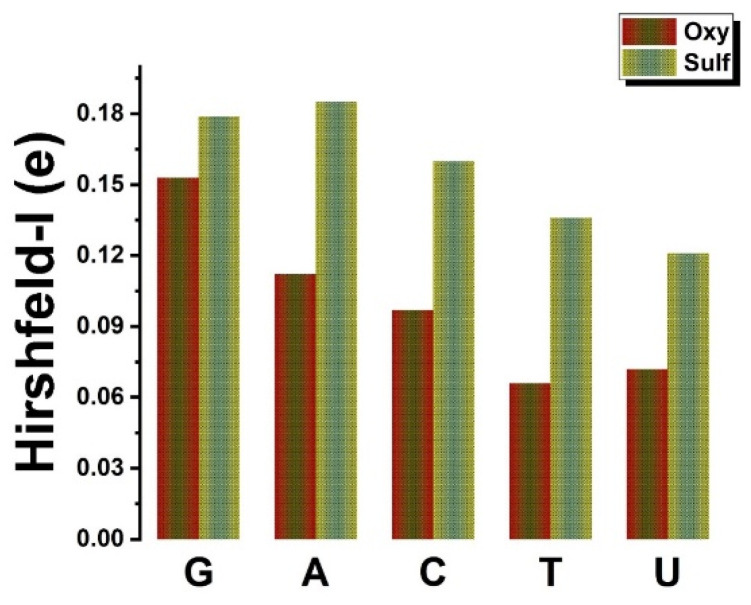
Hirshfeld-I charge of the nucleobase molecules on the Janus MoOS monolayer facets, on O layer and on S layer, respectively.

**Figure 3 biosensors-12-00442-f003:**
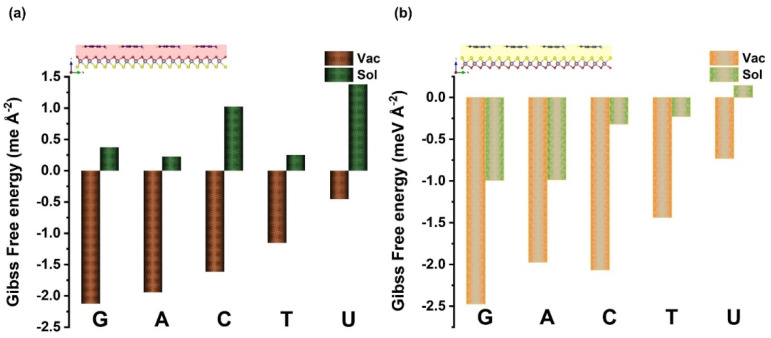
Gibbs free energy (T = 298 K) at vacuum (solvent) of the nucleobases molecules on the Janus MoOS monolayer facets, (**a**) on O layer, (**b**) on S layer, respectively.

**Figure 4 biosensors-12-00442-f004:**
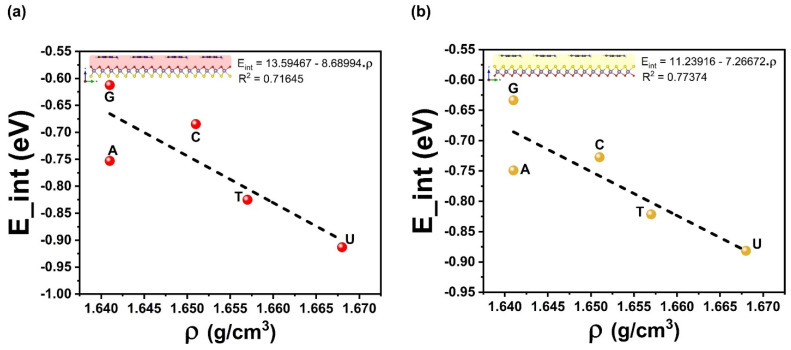
The interaction energy as function of density affinity of the nucleobases molecules on the Janus MoOS monolayer facets, (**a**) on O layer, (**b**) on S layer, respectively.

**Figure 5 biosensors-12-00442-f005:**
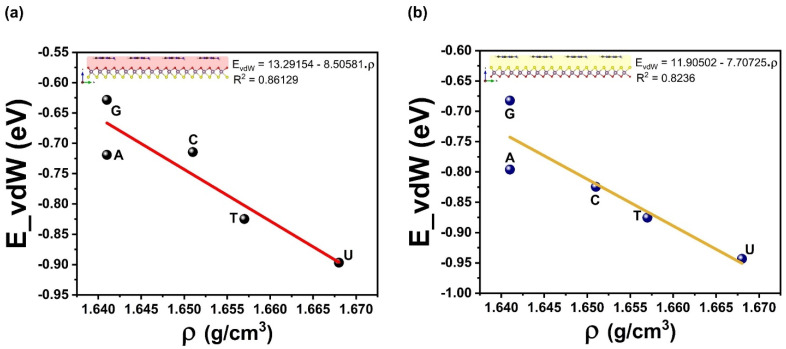
The van der Waals energy as function of density affinity of the nucleobases molecules on the Janus MoOS monolayer facets, (**a**) on O layer, (**b**) on S layer, respectively.

**Figure 6 biosensors-12-00442-f006:**
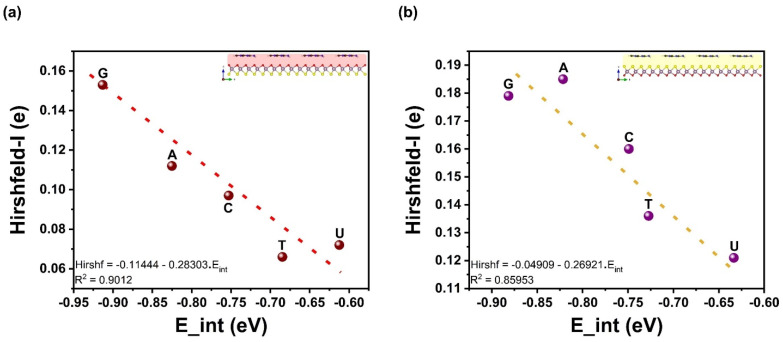
Hirshfeld-I electronic charge as function of interaction energy of the nucleobases molecules on the Janus MoOS monolayer facets, (**a**) on O layer, (**b**) on S layer, respectively.

**Figure 7 biosensors-12-00442-f007:**
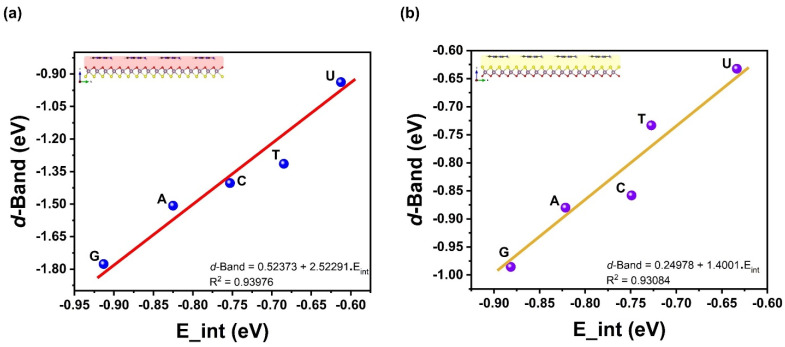
The *d*-band center energy as function of interaction energy of the nucleobases molecules on the Janus MoOS monolayer facets, (**a**) on O layer, (**b**) on S layer, respectively.

**Figure 8 biosensors-12-00442-f008:**
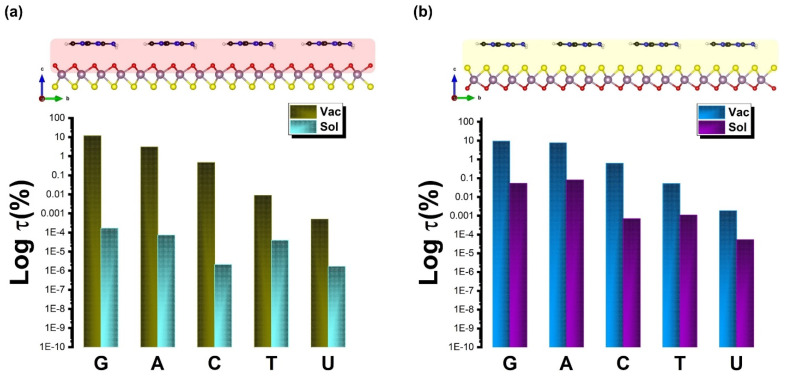
The recovery time of the nucleobases molecules on the Janus MoOS monolayer facets at T = 298 K, (**a**) on O layer, (**b**) on S layer, respectively.

**Figure 9 biosensors-12-00442-f009:**
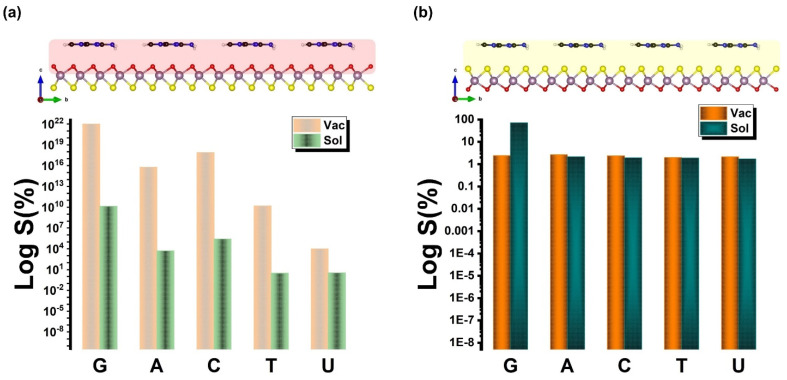
The sensitivity of the nucleobases molecules on the Janus MoOS monolayer facets at T = 298 K, (**a**) on O layer, (**b**) on S layer, respectively.

**Table 1 biosensors-12-00442-t001:** The DNA (RNA) adsorption distance *D*_Ads_ (in Å), adsorption energy *E*_Ads_, interaction energy *E*_Int_, van der Waals energy *E*_vdW_, and covalently energy *E*_Cov_ (All in eV) for dissimilar oxygen (Oxy.) and sulfur (Sul.) layers, respectively.

Ads. State	*D* _Ads_	*E* _Ads_	*E* _Int_	*E* _vdW_	*E* _Cov_
**G–Oxy**	2.77	−0.90	−0.91	−0.89	−0.022
**A–Oxy**	2.80	−0.86	−0.82	−0.82	−0.004
**C–Oxy**	2.74	−0.81	−0.75	−0.72	−0.036
**T–Oxy**	2.94	−0.71	−0.68	−0.71	+0.027
**U–Oxy**	2.85	−0.64	−0.61	−0.63	+0.015
**G–Sul**	3.15	−0.90	−0.88	−0.94	+0.056
**A–Sul**	3.18	−0.89	−0.82	−0.87	+0.050
**C–Sul**	3.06	−0.82	−0.75	−0.79	+0.045
**T–Sul**	3.19	−0.76	−0.73	−0.82	+0.094
**U–Sul**	3.19	−0.67	−0.63	−0.68	+0.047

**Table 2 biosensors-12-00442-t002:** The deformation energy of DNA (RNA) molecules on the Janus MoOS of both oxygen (Oxy.) and sulfur (Sul.) layers, deformation energy of total system *E*_Def__–Tot_, monolayer surface facets *E*_Def__–Lyr_, nucleobases molecule *E*_Def__–Mol_, and covalently energy *E*_Def__–Cov_ (All in eV), respectively.

Ads. State	*E* _Def_ _–_ _Tot_	*E* _Def_ _–_ _Lyr_	*E* _Def_ _–_ _Mol_	*E* _Def_ _–_ _Cov_
**G–Oxy**	+0.016	−0.002	+0.017	+0.021
**A–Oxy**	−0.038	−0.005	−0.033	−0.034
**C–Oxy**	−0.061	−0.017	−0.047	−0.058
**T–Oxy**	−0.027	−0.018	−0.009	−0.024
**U–Oxy**	−0.025	−0.013	−0.013	−0.024
**G–Sul**	−0.011	−0.014	+0.004	−0.005
**A–Sul**	−0.065	−0.018	−0.047	−0.061
**C–Sul**	−0.073	−0.016	−0.056	−0.070
**T–Sul**	−0.030	−0.021	−0.009	−0.027
**U–Sul**	−0.038	−0.023	−0.015	−0.036

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
