# Peer review of "A Peculiar Binding Characterization of DNA (RNA) Nucleobases at MoOS-Based Janus Biosensor: Dissimilar Facets Role on Selectivity and Sensitivity"

_biosensors, 2022, doi:10.3390/bios12070442_

Round 1

Reviewer 1 Report

The manuscript describes the rationales of a Janus MoOS monolayer for its potential as a DNA/RNA sensor. The rationales are supported by the DFT calculation. The results show that the binding occurs through physisorption. The calculation shows good selectivity of the MoOS monolayer between the nucleobase molecules.

Overall, the manuscript is well organized with good flow. Methods are explained in detail. The results are well presented and discussed. Conclusion summarizes the results and discussion.

There are a few awkward sentences that should be revised. Although it is hard to point them out due to no line numbers in the manuscript, they are listed below:

Abstract: “Moreover, the lower …..” This sentence is long and awkward. It should be rephrased.

Page 2: “Sensing is a surface phenomenon …” This sentence should be rephrased.

Page 2: “… with few layers that …” ---> “… with a few layers that …”

Page 2: “… by gate bias.” ---> “… by the gate bias.”

Page 2, bottom: “… have has unveiled crossbreed … on 1H-MoOS …” ---> “… have been unveiled the crossbreed … on a 1H-MoOS …”

Page 3: “All electrons projected …” ---> “All electrons with projected ...”

Page 5: “... onto graphene [52],” ---> “... onto graphene [52].”

Page 7: “with1H-MoOS...” ---> “with 1H-MoOS ...”

Author Response

Abstract: “Moreover, the lower …..” This sentence is long and awkward. It should be rephrased.

Thanks very much to the referee for this important remark, to improve the transparency of the mentioned paragraph we reformulated by the following: “Moreover, the lowering in the adsorption energy leads to an active interaction of the DNA/RNA with the surfaces, accordingly its conduct to shorter the recovery time.”

Page 2: “Sensing is a surface phenomenon …” This sentence should be rephrased.

Thanks so much to the referee for this essential comment. So, to recover the clarity of the above paragraph we restructured the main text by: “Sensing is a surface phenomenon that promote the attraction of two–dimensional (2D) device with contact of the DNA/RNA. Particularly, it leads the possibility of identifying small concentration of molecules with 2D materials [4–6].”

Page 2: “… with few layers that …” ---> “… with a few layers that …”

Many thanks to the referee to spotlight of this statement, so we correct the mentioned sentence.

Page 2: “… by gate bias.” ---> “… by the gate bias.”

Thanks to the referee for pointing out this footnote, we correct the above sentence.

Page 2, bottom: “… have has unveiled crossbreed … on 1H-MoOS …” ---> “… have been unveiled the crossbreed … on a 1H-MoOS …”

Thanks so much to the referee for pointing out this note, we correct the above mentioned phrase.

Page 3: “All electrons projected …” ---> “All electrons with projected ...”

Thanks to the referee for outlined this remark, we correct the above expression.

Page 5: “... onto graphene [52],” ---> “... onto graphene [52].”

Thanks to the referee for pointing out this annotation, we correct the mentioned phrase by the point.

Page 7: “with1H-MoOS...” ---> “with 1H-MoOS ...”

Thanks so much to the referee for sketched this note, we correct the above sentence by space.

Reviewer 2 Report

The manuscript needs checking the grammar and sentences in order to make it more readable before publication.

Author Response

The manuscript needs checking the grammar and sentences in order to make it more readable before publication.

Thanks very much to the referee for drawn out this comment. We actually reformulated carefully the main text to improve the comprehension and the ability of the entire manuscript.

Reviewer 3 Report

This paper is very interesting, written clearly, and I believe quite important and it should be accepted at the current form. Authors should include how the Hirshfeld analysis were studied? Is there any program for that?

Author Response

This paper is very interesting, written clearly, and I believe quite important and it should be accepted at the current form. Authors should include how the Hirshfeld analysis were studied? Is there any program for that?

Many thanks to the referee for this comment. Hirshfeld charge calculations is implemented in VASP 5.4.5. Thereby, during the electronic optimizations the Hirshfeld calculations was integrated self-consistently by using VASP. We utilized VASP program as pre-processing to elucidated Hirshfeld-I charge. We did not used any other script or program for that.

Reviewer 4 Report

This is a numerical study in which the authors show that the introduction of DNA/RNA nucleobases molecule alters the electronic properties of both oxygen and sulfur atomic–layers of the Janus MoOS complex systems.  This finding indicates that dichalcogenides–Janus transition metal may be an adequate biosensor for identifying DNA/RNA bases.

 The subject is timely and of wide interest.

The paper is clear and reports a seemingly well executed and analyzed study. The results, although not validated by the experiment, seem plausible.

 The paper can be accepted after a minor revision:

1. “In the 2D crystals of TMDs, the transition metal atoms are sandwiched between chalcogen atoms forming MX2 composition (M = Mo, W while X = S, Se, Te), and have been fascinating for being atomically ultrathin quantum emitters [8–9].” What do the authors mean here by “quantum emitters”? which phenomenon are they referring to? Be more specific.

2. “Motivated by the great success of graphene, many 2D materials have been developed to serve numerous nanotechnology applications such as environmental bioscience, food, drug vehicle therapy, and biomedicine [10–12].” One of the greatest applications of 2D materials is as constituents of electronic devices. The authors should include electronics in the list of 2D materials applications (see for instance https://doi.org/10.3390/nano8110901 or  https://doi.org/10.1002/aelm.202000094, etc)

 3. “…selectivity, and sensitivity have has unveiled crossbreed for biomolecules on 1H-MoOS Janus system”. Correct the typo.

 4. “Therefore, MoOS monolayer is primly appropriate for physisorption of nucleobases than graphene, tellurene, phosphorene and boron–nitride nanosheets.” Based on the results shown, I agree with this conclusion. However, I wonder if the authors have considered the presence of defects such as O or S vacancies and how they affect the physisorption properties of MoOS.

Author Response

  1. “In the 2D crystals of TMDs, the transition metal atoms are sandwiched between chalcogen atoms forming MX2 composition (M = Mo, W while X = S, Se, Te), and have been fascinating for being atomically ultrathin quantum emitters [8–9].” What do the authors mean here by “quantum emitters”? which phenomenon are they referring to? Be more specific.

We would like to thank the reviewer for pointing out this outline. Based on the paper published by Huang et al. and Wang et al. it seems that the chemisorption of NO2 on MoS2 as sensor is attractive with high brightness, and the dichalcogenides 2D materials display the capability to be integrated with a high range of electronic and photonic platforms.

  1. “Motivated by the great success of graphene, many 2D materials have been developed to serve numerous nanotechnology applications such as environmental bioscience, food, drug vehicle therapy, and biomedicine [10–12].” One of the greatest applications of 2D materials is as constituents of electronic devices. The authors should include electronics in the list of 2D materials applications (see for instance https://doi.org/10.3390/nano8110901 or  https://doi.org/10.1002/aelm.202000094, etc)

We would like to thank the reviewer for the comment above; we correct the introduction by adding the “…electronic device…” interest on the text with the recommended references.

  1. “…selectivity, and sensitivity have has unveiled crossbreed for biomolecules on 1H-MoOS Janus system”. Correct the typo.

Many thanks to the referee for this intriguing comment. We do correct the sentence by “…selectivity, and sensitivity have been unveiled crossbreed for biomolecules on 1H-MoOS Janus system”.

  1. “Therefore, MoOS monolayer is primly appropriate for physisorption of nucleobases than graphene, tellurene, phosphorene and boron–nitride nanosheets.” Based on the results shown, I agree with this conclusion. However, I wonder if the authors have considered the presence of defects such as O or S vacancies and how they affect the physisorption properties of MoOS.

Thanks so much to the referee for the tips. By creating atomic vacancies, we believe that the electronic charge distributions of the surfaces will change and conduct the creation of new chemical bond between the biomolecules and the Janus 1H-MoOS system that is initiate by the unsaturated charge near of the surfaces. This is going to be a follow-up paper to uncover different chemical aspect and properties such as to see the effect of the surface defects, molecular self-assembly, MD simulations ….etc.